# Indistinguishable photons from an artificial atom in silicon photonics

Lukasz Komza[1,2,7], Polnop Samutpraphoot[2,3,7], Mutasem Odeh[2,3], Yu-Lung Tang[1,2], Milena Mathew[2,3], Jiu Chang[3], Hanbin Song[4], Myung-Ki Kim[3,5], Yihuang Xiong[6], Geoffroy Hautier[6] & Alp Sipahigil[1,2,3] ✉

Silicon is the ideal material for building electronic and photonic circuits at scale. Integrated photonic quantum technologies in silicon offer a promising path to scaling by leveraging advanced semiconductor manufacturing and integration capabilities. However, the lack of deterministic quantum light sources and strong photon-photon interactions in silicon poses a challenge to scalability. In this work, we demonstrate an indistinguishable photon source in silicon photonics based on an artificial atom. We show that a G center in a silicon waveguide can generate high-purity telecom-band single photons. We perform high-resolution spectroscopy and time-delayed two-photon interference to demonstrate the indistinguishability of single photons emitted from a G center in a silicon waveguide. Our results show that artificial atoms in silicon photonics can source single photons suitable for photonic quantum networks and processors.

Silicon quantum technologies based on integrated photonics[1] offer a promising path to scaling by leveraging advanced semiconductor manufacturing and integration capabilities[2,3]. Current approaches to fault-tolerant photonic quantum computation use weak material nonlinearities and measurements to probabilistically generate photon pairs and implement two-qubit gates[1,4]. The lack of deterministic quantum light sources[5], photon-photon gates[6], and quantum memories[7] in silicon photonics poses a major challenge to scalability and requires very large resource overheads[8]. Coherently controlled quantum emitters in a reconfigurable photonic circuit can enable hardware-efficient universal quantum computation[9,10] and time-multiplexed quantum networking[11]. Silicon photonics provides a mature platform for low-loss reconfigurable integrated photonics[12]. However, an atomic source of indistinguishable photons in silicon has been missing[13]. We address this challenge by demonstrating telecom-band indistinguishable photon generation from an artificial atom in silicon photonics.

Artificial atoms in solids enable single-photon level optical nonlinearities for realizing deterministic single-photon sources, as well as two-photon gates and long-range spin-spin entanglement[7,14–16]. While defect-based photoluminescence (PL) in silicon has been studied for decades[17,18], bright telecom-band single-photon emission from a broad diversity of artificial atoms in silicon was only recently shown[19–23]. In order for silicon artificial atoms to function as quantum-coherent light sources, their emission has to satisfy spatiotemporal indistinguishability[24]. In this work, we integrate a silicon color center into a photonic waveguide, show pulsed single-photon generation, and demonstrate that successive photons emitted are indistinguishable.

## Results

### An artificial atom in a silicon waveguide

Our device consists of a G center created in a silicon photonic waveguide (Fig. 1). The G center is a complex defect in silicon that consists of two substitutional carbon ($C_S$) atoms and an interstitial silicon ($S_i$). It

¹Department of Physics, University of California, Berkeley, Berkeley, CA 94720, USA. ²Materials Sciences Division, Lawrence Berkeley National Laboratory, Berkeley, CA 94720, USA. ³Department of Electrical Engineering and Computer Sciences, University of California, Berkeley, Berkeley, CA 94720, USA. ⁴Department of Materials Science and Engineering, University of California, Berkeley, Berkeley, CA 94720, USA. ⁵KU-KIST Graduate School of Converging Science and Technology, Korea University, Seoul 02841, Republic of Korea. ⁶Thayer School of Engineering, Dartmouth College, 14 Engineering Dr, Hanover, NH 03755, USA. ⁷These authors contributed equally: Lukasz Komza, Polnop Samutpraphoot. ✉e-mail: alp@berkeley.edu

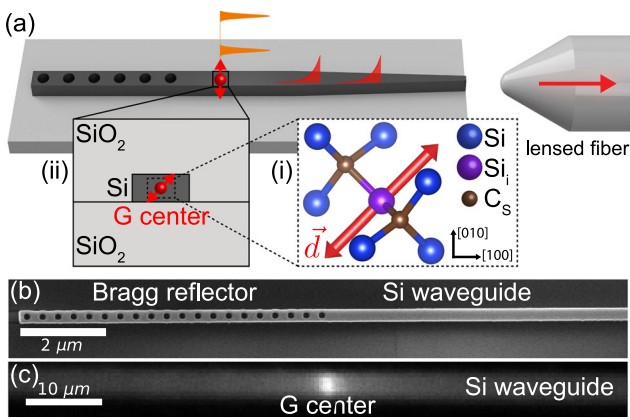

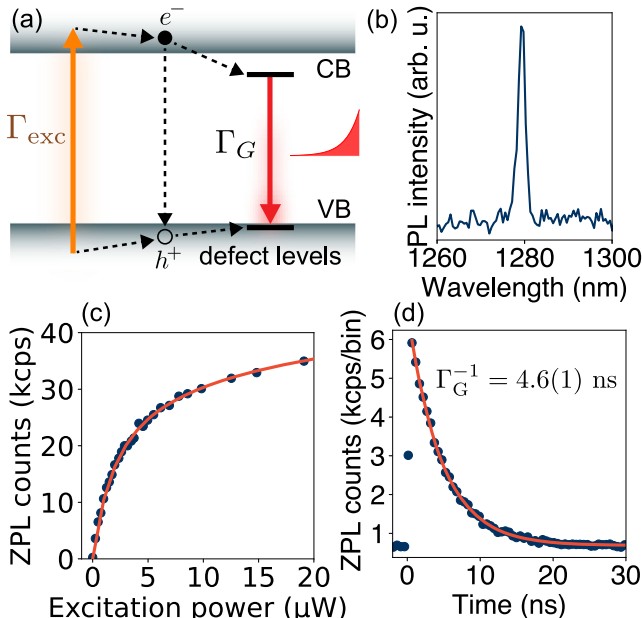

**Fig. 1 | An atomic quantum light source in silicon photonics. a** Device and measurement schematic. (i) A G center is created inside a silicon photonic waveguide via ion implantation. (ii) Silicon waveguide cross section. G center transition dipole $\vec{d}$ is along ⟨110⟩. Laser excitation (orange) of the G center results in single-photon emission into the waveguide (red) which is collected using a lensed fiber. **b** Scanning electron micrograph of the photonic waveguide near the broadband Bragg reflector. **c** PL image of the waveguide shows bright emission from an isolated G center.

**Fig. 2 | Photodynamics of a G center in a waveguide. a** Above-bandgap excitation ($\Gamma_{exc}$) creates excess carriers in the conduction and valence bands (CB, VB) which recombine at the localized defect levels ($\Gamma_G$) and produce single-photon emission. **b** PL spectrum of the ZPL of the G center. **c** Excitation power dependence of the ZPL PL shows saturated emission. **d** PL lifetime measurement with pulsed excitation.

emits in the telecommunication O-band with a zero-phonon line (ZPL) at 1278 nm[17,18]. We create G centers inside the 220 nm device layer of a silicon-on-insulator (SOI) wafer by $^{12}$C ion implantation at 36 keV and a fluence of $10^{12}$ cm$^{-2}$, followed by rapid thermal annealing at 1000°C for 20 seconds. Following device fabrication, these parameters result in the creation of approximately one G center in a 100 $\mu$m-long waveguide, allowing investigation of a single spatially isolated defect. The creation yield of G centers can be dramatically increased via proton irradiation[25] or by alternative fabrication steps as discussed in Supplementary Information (SI) Note 2.

Upon above-bandgap excitation, the G center emits photons via the radiative recombination of electron-hole pairs at localized defect levels (Fig. 2a). The dipole emission is guided by a single-mode silicon waveguide coupled to a single-mode lensed fiber with 50% efficiency using an adiabatic mode converter (Fig. 1a, Fig. S3). The waveguide is terminated with a Bragg reflector for single-sided measurements and coupling efficiency calibration. The collected photons are detected using a spectrometer (Fig. 2b) or superconducting nanowire single-photon detectors (SNSPD) with a quantum efficiency of 60%. The sample is housed in a cryostat and measured at 3.4 K. Materials, fabrication, photonic design, setup, and first principles calculation details are provided in the Methods and SI.

### Optical properties of a G center in a waveguide

To locate a single G center, we spatially scan a free-space excitation beam at 635 nm and detect photons emitted into the waveguide through the lensed fiber. Figure 1c shows the resulting PL image of the waveguide where we observe an isolated emitter with a measured photon rate of 18 kHz using a bandpass filter (1280 ± 6 nm) centered at the G center ZPL (emission spectrum shown in Fig. 2b). In the following experiments, we probe the linear and nonlinear optical responses of this G center using time- and spectrally- resolved single-photon detection. We study the saturation response of the G center by measuring the power dependence of the ZPL emission rate $R_{ZPL}$ on the excitation power $P$. The power dependence is modeled by $R_{ZPL} = R_{sat}P/(P_{sat} + P) + \alpha P$, where the two terms correspond to a two-level atomic response and a weak linear background. The fit yields a saturated count rate of $R_{sat} = 35$ kcps and a saturation power $P_{sat} = 2.4 \mu$W (Fig. 2c). Next, we use a pulsed laser at 705 nm to measure the PL lifetime of the emitter to be $\Gamma_G^{-1} = 4.6(1)$ ns (Fig. 2d). For each excitation pulse, we detect a ZPL photon with a probability of $0.4(1) \times 10^{-3}$ (SI Note 5). We calibrate the losses in our setup

and use the ZPL branching ratio of 0.18 to estimate the probability the G center emits into the waveguide $\beta = \Gamma_{1D}/\Gamma_G = 0.014$, where $\Gamma_{1D}$ is the radiative emission rate into the waveguide. We estimate a radiative lifetime upper bound of 260 ns. Our first principles calculations in SI Note 4 predict a radiative lifetime of 225(75) ns.

### Optical coherence

We probe the optical coherence of the G center by measuring the ZPL emission spectrum using a tunable Fabry-Perot (FP) cavity with a linewidth of $\kappa_{FP}/2\pi = 3.4$ GHz (Fig. 3a). The resulting spectrum, which is a convolution of the ZPL emission and the FP transmission, shows a total linewidth of 6.2 GHz. After deconvolving the cavity response, we find the G center emission linewidth to be $\Gamma/2\pi = 2.8$ GHz (Fig. 3b). Next, we characterize the photon statistics of the G center emission by measuring the normalized intensity correlations $g^{(2)}(\tau)$ under pulsed 705 nm excitation at a repetition period $\Delta\tau = 25$ ns (Fig. 2c). We observe antibunched intensity correlations at zero delay $g^{(2)}(0) = 0.15(2) < 0.5$ which confirm single-photon emission. The $g^{(2)}(0)$ value is limited by contributions from exponential tails originating from the ratio of the repetition period and the excited state lifetime ($\Delta\tau/\Gamma_G^{-1}$), imperfect extinction in pulsed laser downsampling, and dark counts (SI Note 6). We benchmark the long-term stability of the G center emission by analyzing the intensity correlations up to seconds of delays under CW excitation. The results in Fig. 3d, e show a flat response, indicating stable single-photon emission without any excess intensity fluctuations for $\tau > 50$ ns. We observe bunching at shorter timescales which has been attributed to the presence of a metastable state[19,22].

### Time-resolved two-photon quantum interference

Photon indistinguishability requires a high degree of spatio-temporal overlap between single-photon wavepackets[24]. We use a time-delayed Hong-Ou-Mandel (HOM) interference experiment to test the indistinguishability of successive single-photon pulses from the G center[26,27]. We interfere successive single photons (red and blue pulses in Fig. 4a) using a fiber-based time-delayed Mach-Zehnder interferometer (MZI) where one path has an additional delay $\Delta\tau = 25$ ns,

matched to the laser repetition period. We adjust the relative polarization between the two MZI paths to control the mode overlap and photonic indistinguishability at the second beam splitter.

The results of HOM interference between parallel and orthogonally polarized single-photon pairs from a G center are shown in Fig. 4b, c. When the polarizations of the two interfering photons are parallel (red data, indistinguishable case), we see the characteristic HOM dip[28] resulting from two-photon quantum interference at short time delays with $g_{\parallel}^{(2)}(0) = 0.26(4) < 0.5$. When we tune the photons to be orthogonally polarized so that they are intentionally distinguishable (blue data), the HOM dip disappears and we obtain $g_{\perp}^{(2)}(0) = 0.69(5)$. A comparison of the normalized coincidence probability at zero time delay yields an HOM interference visibility of $\chi(0) = 1 - g_{\parallel}^{(2)}(0)/g_{\perp}^{(2)}(0) = 0.62(4)$. We identify a 0.37(18) ns window over which $\chi(\tau) > 0.5$. The integrated visibility over the entire

wavepacket is 0.004, which can be improved to close to 1 with future advances discussed below.

The temporal dynamics of the experimentally measured $g_{\parallel}^{(2)}(\tau)$ can be described using the formula $G^{(2)}(\tau) = \frac{e^{-\tau/T_1}}{4T_1}(1 - \chi e^{-(\Gamma^{HOM}\tau)})$ which is derived in SI Note 6. In the model, we use the value of $T_1 = 4.6(1)$ ns from the lifetime measurements in Fig. 3. $\Gamma_{HOM}$ describes an effective linewidth of the emitter which represents the average detuning between successive single photon pulses separated by 25ns. We find $(\Gamma^{HOM})^{-1} = 0.4 \pm 0.1$ ns, corresponding to an effective short-timescale linewidth of $\Gamma^{HOM}/2\pi = 0.4(1)$ GHz. We note that this effective linewidth is about an order of magnitude smaller than the measured linewidth of the emission spectrum in Fig. 3, and filtering effects from the FP cavity have a negligible impact (2% reduction) on $\Gamma^{HOM}$. We attribute the smaller effective linewidth to the time dynamics of spectral diffusion in the system. Fig. 3b was acquired over a ten-minute period, whereas we interfere two subsequently emitted photons separated by 25 ns in our HOM experiment. These results show that the emission spectrum is stable over short timescales, a feature favorable for on-chip multi-photon experiments. The zero-delay visibility is limited by the timing jitter of our detector pair (250 ps), imperfect polarization overlap, finite lifetime-to-repetition period ratio, and dark counts (SI Note 6).

## Discussion

Our results show that color centers in silicon can generate indistinguishable photons at the telecom-band in silicon photonics. This experiment is enabled by the large transition dipole moment ($2.8 \pm 0.5$ Debye calculated from first principles in SI Note 4), the optical coherence of the G center, and efficient collection of single photons from the G center using silicon photonics. In the following, we discuss open questions and approaches to use this platform to develop high-purity waveguide-integrated single photon sources in silicon.

Photonic indistinguishability is a prerequisite for achieving strong photon-photon interactions[15] and building large photonic quantum states[29,30]. The primary limitation of our results is the small temporal window over which indistinguishable photons may be post-selected, caused by optical decoherence of the G center. To relax the post-selection requirement, the decoherence time scale must be sufficiently long when compared to the excited state lifetime. This can be achieved by reducing the excited state lifetime and improving the optical coherence of the emitter. The coherent interaction rate between the emitter and photons can be enhanced in high Q/V silicon photonic resonators[31–34] with mode volumes $V \sim 0.1\lambda^3$ and quality factors $Q \sim 10^6$. Resonant excitation, dynamic stabilization[35], and embedding emitters in p-i-n junctions can suppress charge noise[36], reduce spectral diffusion, and improve optical coherence. Additionally, the discovery of

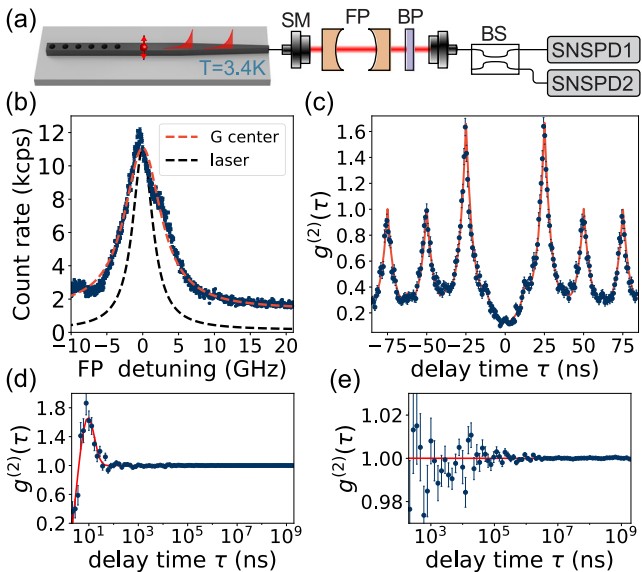

**Fig. 3 | Quantum coherence of G center emission. a** Emission from the G center is collected through a single mode (SM) fiber and analyzed with a tunable Fabry Perot (FP) cavity. BP: Bandpass filter, BS: beamsplitter. **b** Measured emission linewidth (red fit): 6.2(1) GHz, FP linewidth (black, using reference laser): 3.4(1) GHz. Calculated G center linewidth after deconvolving the FP response: 2.8(1) GHz. **c** Normalized intensity correlations at the detectors $g^{(2)}(\tau)$ show single-photon emission $g^{(2)}(0) = 0.15(2) < 0.5$. **d, e** Long term intensity correlations under continuous wave excitation show stable single-photon emission. **c–e** Error bars represent Poissonian noise, and the error of reported values is the error of the fits.

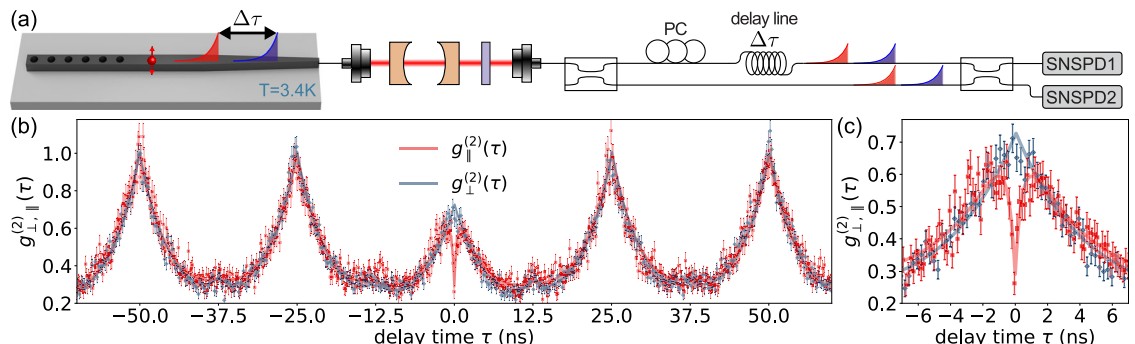

**Fig. 4 | Quantum interference of single photons from a G center in silicon photonics. a** Successive photons ($\Delta\tau = 25$ ns) interfere via a time-delayed Mach-Zehnder interferometer. Indistinguishability of the interfering single photons is adjusted by setting their relative polarizations using a polarization controller (PC). **b** Two-photon interference measurement. $g^{(2)}(\tau)$ for orthogonal (blue) and parallel

(red) polarizations. **(c)** When the two photons are indistinguishable, quantum interference results in the antibunching dip at short time delays. We observe nearly identical correlations outside of the quantum interference window. **b, c** Error bars represent Poissonian noise, and errors in $g^{(2)}(\tau)$ values are extracted from the error of the fits.

new centrosymmetric artificial atoms[21] in silicon will make these systems more robust against spectral diffusion[37]. With such improvements, we estimate that G centers in high Q/V cavities can reach cavity-QED parameters of $\{g, \kappa, \gamma\}/2\pi \approx \{1.3, 0.25, 0.2\}$ GHz and enter the strong-coupling regime with a cooperativity $C = 4g^2/\kappa\gamma \sim 100$. For indistinguishable photon generation in the Purcell regime, the cavity external decay rate can be intentionally increased to achieve high efficiency and high indistinguishability[5].

Emerging silicon artificial atoms with electron and nuclear spins[23] can also be introduced to this device platform to implement quantum processor and repeater building blocks, as was shown in other material platforms[7,16,38,39]. The realization of such high-fidelity spin-photon gates with silicon artificial atoms will open up the possibility of scaling spin[2] and photonic[1] quantum processors and repeaters using advanced CMOS manufacturing and integration capabilities[2,3,13].

## Methods

### Device fabrication and design

The sample was prepared from a 1 cm × 1 cm chip diced from a 200 mm, high-resistivity (Float-zone, ≥3000 Ωcm), 220 nm SOI wafer prepared using the SmartCut method. The fabrication process is summarized in Fig. S2. The backside of the chip was partially diced to enable cleaving of the chip at desired locations to expose the waveguide facet for fiber coupling. Carbon implantation and high temperature annealing were carried out before any lithography steps. An etch mask for photonic structures was defined through electron beam lithography on HSQ resist. The mask was developed using a NaOH/NaCl developer chemistry. The 220 nm device layer was etched in a $Cl_2$/HBr chemistry. Finally, approximately 3 $\mu$m of silicon oxide cladding was deposited on the surface of the chip through PECVD. The resulting waveguides have the cross section shown in Fig. S3. We used Finite Element Method (FEM) and Finite-Difference Time-Domain (FDTD) solvers to design a Bragg reflector for the fundamental TE mode based on a photonic crystal with lattice constant of 370 nm and ellipsoid holes with the principle axes of {170, 200} nm. This design yields near-unity reflection for TE polarization over a 150 nm band centered at 1330 nm. The center of the band was biased towards wavelengths longer than the G center ZPL to include the phonon sideband. The waveguide width of 300 nm was optimized to achieve the maximum electric field intensity at the center of the waveguide, and therefore the coupling strength to the emitter. The waveguide width was tapered down to 130 nm over 50 $\mu$m for maximum coupling to a 2.5 $\mu$m lensed fiber (Fig. S3). The waveguide is fabricated along the $\langle 100 \rangle$ crystal axis.

## Data availability

The data generated in this study are available in the Zenodo database at https://doi.org/10.5281/zenodo.11553484.

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

## Acknowledgements

We thank Quantum Opus for custom SNSPD installation, Auden Young, Andrew Kim, and Xudong Li for technical assistance, and Zihuai Zhang for feedback on the manuscript. The devices used in this work were fabricated at UC Berkeley's NanoLab. L.K. and A.S. acknowledge funding from the NSF QuIC-TAQS program through award No 2137645. P.S., H.S., Y.X., and G.H. acknowledge funding from the U.S. Department of Energy, Office of Science, Basic Energy Sciences in Quantum Information Science under Award Number DE-SC0022289 for materials processing and first principles modeling. M.O. acknowledges funding from the NSF Challenge Institute for Quantum Computation (CIQC) under award OMA-2016245 for device fabrication. This research used resources of the National Energy Research Scientific Computing Center, a DOE Office of Science User Facility supported by the Office of Science of the U.S. Department of Energy under Contract No. DE-AC02-05CH11231 using NERSC award BES-ERCAP0020966.

## Author contributions

L.K. and P.S. built the experimental setup, performed the measurements, and analyzed data. M.O. and L.K. fabricated the devices. Y.-L. T., M.M., J. C., H. S., and M.K. assisted with experiments and analysis. Y.X. and G.H. performed first-principles calculations. P.S., L.K. and A.S. wrote the manuscript with input from all authors. A.S. conceived and supervised the work. All authors discussed the results and analysis.

## Competing interests

The authors declare no competing interests.
