## [Peer Review File · Nature Communications]

Indistinguishable photons from an artificial atom in silicon photonicsEditorial note: This manuscript has been previously reviewed at another journal that is not operating a transparent peer review scheme. This document only contains reviewer comments and rebuttal letters for versions considered at *Nature Communications*.

REVIEWER COMMENTS

Reviewer #1 (Remarks to the Author):

The revised manuscript by Komza et al. effectively addresses the concerns raised during the previous review. The authors have made significant improvements in presenting their work, addressing all the issues previously highlighted. Furthermore, by adopting a more measured approach in discussing topics such as spin-photon entanglement, the revised manuscript accurately conveys the achievements of the research.

While it is disappointing that the study lacks spin-dependent photoemission and below-bandgap excitation, it is important to recognize that the work represents a significant advancement in the development of quantum devices based on isolated defect systems in silicon. This is particularly noteworthy given the preference for silicon in modern industry applications. Therefore, I highly recommend the publication of this manuscript in *Nature Communications*.

Reviewer #2 (Remarks to the Author):

On the whole, the authors have followed the referees' comments and applied the necessary corrections to their manuscript. In particular, they have explained and discussed the importance of temporal filtering and added further information on sample fabrication and defect creation. Following remarks from Referee #1, they have also improved their manuscript by removing previous references to spin-photon interface as the G center optical transition occurs between spin singlet states. This does not reduce the scope of the results since even without spin memory, developing deterministic sources of indistinguishable single photons in silicon would be a game changer. With the exception of a few issues detailed below, the answers to the questions are generally satisfactory.

1- Comment to Reply & Revision 2.1, & Reply 2.6 (lines 128-129): It would be more convincing to provide an estimate of the expected linewidth with such high quality factor and small mode volume and compare it to the linewidths measured in the current work, rather than letting the readers do their own calculations.

In addition, the authors should cite recent references on Purcell-enhancement of the G center emission:

- for ensemble of G centers: Lefaucher et al., Applied Physics Letters 122 (2023),
- for single G center: Saggio et al., 10.48550/arXiv.2302.10230 (2023).

2- Comment to Reply 2.7: I believe the authors' answer is incorrect, as well as the integration between Equations (3) and (4) in the Methods (lines 310-311). According to their Reply 3.7, the authors have followed the analysis from Lettow et al., PRL 104 (2010) that has been developed for 2 emitters that have a non-zero average frequency detuning resulting in a beating observed in the HOM dip. Here there is a single emitter, so the frequency difference between the first photon emitted at t_0 and the second emitted at $t_0 + \tau$ depends on t_0 due to environment fluctuations (as indicated on lines 325-326). In other terms, the integration over t_0 for the random variable $\Delta(t_0) = \omega_2(t_0 + \tau) - \omega_1(t_0)$ is missing there. As a consequence, the sentence on lines 312-322 "However, large detunings result in very rapid oscillations that cannot be measured to the detector and electronics timing jitters" does not make sense. This must be corrected. Later, there is also an other Δ_{12} that is introduced but not defined.

3- Comment to Reply 1.2: I agree with Referee #1 that the authors' claiming "that the single photon source in this report is coherent" is questionable. The last sentence from the abstract is ambiguous (lines 10-12): "Our results show that artificial atoms in silicon photonics can source highly coherent single photons suitable for photonic quantum networks and processors". The concept of optical coherence is very broad and can take different meanings. The authors' reference to "coherent single-photon source" gives the false impression of relating to the higher-order photon coherence required to probe the photon indistinguishability using two-photon quantum interference. Given the significant inhomogeneous broadenings reported here and the strong temporal filtering needed, this specific photon coherence is rather bad in the current work. Instead, the authors explain in

Reply 1.2 that they are referring to a mixture of "first-order coherence" associated to "narrow linewidth" and "second-order coherence" linked to "single-photon statistics". First I am not convinced that the linewidths are narrow here (especially compared to the lifetime-limited linewidth). Secondly this first-order coherence is related to single-photon interference, which is to be avoided when implementing HOM (as in the current manuscript, the interferometer optical path difference being much longer than the single photon coherence length). At last, the $g^{(2)}$ function is the second-order autocorrelation function. Here it seems inappropriate to refer to it as a probe for second-order coherence as neither coherence time nor frequency linewidth are extracted. In my opinion, the term "coherent single-photon source" brings confusion here, which tends to distract the readers from the main result (first HOM on a single defect in silicon). I would suggest that the authors clarify or even change this term.

4- Methods line 290: The authors compare the estimate they get for the G center radiative lifetime to a previous experiment (Ref. 19) from which a much shorter lifetime is extracted. According to an arxiv preprint recently out (Durand et al., 10.48550/arXiv.2402.07705 (2024)), the defect investigated in Ref. 19 is not a G center but another point defect. This comparison is therefore no longer relevant, but it is likely that the authors were not aware of these findings at the time of their manuscript resubmission.

Admittedly, photon indistinguishability has been demonstrated here over a very short timescale and with significant temporal filtering. Nevertheless, this difficult HOM experiment has been performed on a new quantum system, the G center in silicon, that is still little known due to its recent isolation (Baron et al., ACS Photonics 9 (2022), Hollenbach et al., Nat. Comm. 13 (2022)), and integrated inside a SOI waveguide. As replied by the authors, these findings are a first important step in the control of the photon emission of single artificial atoms in silicon and there is plenty of room for improvements. This work will foster new investigations by many groups in the scientific community. Provided the above comments are addressed in a second revised manuscript, I recommend publication of these results in Nature Communications journal.

Reviewer #3 (Remarks to the Author):

The authors have not made significant changes to the manuscript in their revision. In particular, a more detailed discussion of the observed indistinguishability is still lacking in the main text.

All referees had critical questions on the analysis; while responses are given, only slight changes have been made to the manuscript. Assessing the indistinguishability based solely on $g^2(0)$ remains incomplete, and the brief treatment of the observed time constant of the quantum interference dip in the main text leaves questions open. One must study the supplemental material to obtain a more detailed analysis and a discussion of this result. It would be helpful to have more of this in the main text. I would have expected a mentioning of the indistinguishability of the entire photon wave packet for comparison, as well as a discussion of how the results compare to the measured spectrum.

I now also wonder what the effect of the filter FP cavity is, given that it is more narrow than the emitter linewidth – I missed this aspect when reviewing the first version of the manuscript. What indistinguishability would be observable without the filter cavity? A narrow enough spectral filter can produce indistinguishable photons from any single photon emitter, at the cost of low efficiency. How is this trade-off here? While the effective linewidth for quantum interference derived in the supplement suggests that the filter effect of the cavity may be small, an estimate should be given.

Overall, I think that these aspects should be addressed before publication.

We are pleased to submit the revised version of our manuscript “Indistinguishable photons from an artificial atom in silicon photonics” for consideration by *Nature Communications*. We thank the reviewers for their valuable input. In the following, we address their feedback point-by-point and include revisions in the manuscript where appropriate. A version of the revised manuscript where the changes are highlighted is included in the resubmission.

We believe that the revised manuscript appropriately addresses all questions raised by the referees, has significantly improved in clarity, and meets the high standards for publication in *Nature Communications*.

Referee #1

The revised manuscript by Komza et al. effectively addresses the concerns raised during the previous review. The authors have made significant improvements in presenting their work, addressing all the issues previously highlighted. Furthermore, by adopting a more measured approach in discussing topics such as spin-photon entanglement, the revised manuscript accurately conveys the achievements of the research.

While it is disappointing that the study lacks spin-dependent photoemission and below-bandgap excitation, it is important to recognize that the work represents a significant advancement in the development of quantum devices based on isolated defect systems in silicon. This is particularly noteworthy given the preference for silicon in modern industry applications. Therefore, I highly recommend the publication of this manuscript in *Nature Communications*.

Reply (1.1) We thank the referee for their favorable review of our work.

Referee #2

On the whole, the authors have followed the referees’ comments and applied the necessary corrections to their manuscript. In particular, they have explained and discussed the importance of temporal filtering and added further information on sample fabrication and defect creation. Following remarks from Referee #1, they have also improved their manuscript by removing previous references to spin-photon interface as the G center optical transition occurs between spin singlet states. This does not reduce the scope of the results since even without spin memory, developing deterministic sources of indistinguishable single photons in silicon would be a game changer. With the exception of a few issues detailed below, the answers to the questions are generally satisfactory.

Reply (2.1) We thank the referee for the favorable summary of our work.

1-Comment to Reply & Revision 2.1, & Reply 2.6 (lines 128-129): It would be more convincing to provide an estimate of the expected linewidth with such high quality factor and small mode volume and compare it to the linewidths measured in the current work, rather than letting the readers do their own calculations.

Reply (2.2) We thank the referee for raising this point. We now provide the estimated parameters directly in the text. A detailed discussion explaining the changes is provided below.

In the original text, we referred to an estimated Purcell factor of $\sim 10^6$ calculated from a small cavity mode volume

$V \sim 0.1\lambda^3$ and large cavity Q factor $Q \sim 10^6$. In the revised version, we directly provide the estimated cavity-QED parameters based on such realistic improvements discussed in the text: $\{g, ic, \gamma\}/2\pi \sim \{1.3, 0.25, 0.2\}$ GHz. We also correct that these parameters would put the system in the strong coupling regime instead of the Purcell regime of cavity QED with a cooperativity of $C = 4g^2/ic\gamma \sim 100$. We have changed the wording to accurately describe this, and note that operation in the Purcell ($ic > \gamma, g$) regime can be achieved by deliberately choosing larger ic values while still achieving high photon indistinguishability (i.e. $C > 1$). We note that the above estimate takes into account the ZPL branching ratio, and the quantum efficiency estimates discussed in the text. We use a dipole moment for the ZPL transition of 1.3 Debye based on parameters discussed in the efficiency analysis section of the Methods.

Revision (2.2) We provided the estimated cavity QED values in the main text.

Line-indexed revisions: 131, 136-140

In addition, the authors should cite recent references on Purcell-enhancement of the G center emission: - for ensemble of G centers: Lefaucher et al., Applied Physics Letters 122 (2023), - for single G center: Saggio et al., 10.48550/arXiv.2302.10230 (2023).

Revision (2.3) We have added citations to recent cavity-enhanced results.

Line-indexed revisions: 132

2- Comment to Reply 2.7: I believe the authors' answer is incorrect, as well as the integration between Equations (3) and (4) in the Methods (lines 310-311). According to their Reply 3.7, the authors have followed the analysis from Lettow et al., PRL 104 (2010) that has been developed for 2 emitters that have a non-zero average frequency detuning resulting in a beating observed in the HOM dip.

Here there is a single emitter, so the frequency difference between the first photon emitted at t_0 and the second emitted at $t_0 + \tau$ depends on t_0 due to environment fluctuations (as indicated on lines 325-326). In other terms, the integration over t_0 for the random variable $\Delta(t_0) = \omega_2(t_0 + \tau) - \omega_1(t_0)$ is missing there. As a consequence, the sentence on lines 312-322 "However, large detunings result in very rapid oscillations that cannot be measured to the detector and electronics timing jitters" does not make sense. This must be corrected.

We agree with the referee that our analysis leading to equations (3) and (4) describes two photon interference where the two photons have a fixed detuning Δ_{12} . In our analysis, we have chosen to start from this simple case and calculate the joint detection probability for a fixed detuning between the two photons interfering (following Lettow). We note that for the case where Δ_{12} is assumed to be fixed, the integration between Eq (3) and (4) is correct.

As the referee points out and is discussed in the text, the detuning is not a constant but evolves in time due to spectral diffusion. A complete description of the correlation properties of the detuning variable ($\langle \Delta_{12}(t_0)\Delta_{12}(t_0 + \tau) \rangle$) is beyond the scope of this work, and would require substantial additional studies.

For this reason, in Eqn. (9) we use the simplest model to describe the experiments using a quasistatic approximation. In other words, this model assumes that for each emission event, there is a well defined emission frequency, and the relative detuning between successive events is sampled from a Lorentzian distribution given by Eqn. (8). This allows us to use the joint probability distribution calculated in Eqn. (5), and classically average over an effective probability distribution given by Eqn. (8) to reach the result in Eqn. (10). This classical averaging allows

us to model the experimentally observed exponential decay.

Revision (2.5) We have revised the time-resolved two-photon interference section of the supplementary information to clarify the points raised above. We also added a description of the model in the main text.

Line-indexed revisions: 297-299, 304-305, 332, 104-109

Later, there is also an other Δ_{12} that is introduced but not defined.

Revision (2.6) We have defined Δ_{12} as the relative detuning between successive pulses. *Line-indexed revisions:* 324

3-Comment to Reply 1.2: I agree with Referee #1 that the authors' claiming "that the single photon source in this report is coherent" is questionable. The last sentence from the abstract is ambiguous (lines 10-12): "Our results show that artificial atoms in silicon photonics can source highly coherent single photons suitable for photonic quantum networks and processors". The concept of optical coherence is very broad and can take different meanings. The authors' reference to "coherent single-photon source" gives the false impression of relating to the higher-order photon coherence required to probe the photon indistinguishability using two-photon quantum interference. Given the significant inhomogeneous broadenings reported here and the strong temporal filtering needed, this specific photon coherence is rather bad in the current work. Instead, the authors explain in Reply 1.2 that they are referring to a mixture of "first-order coherence" associated to "narrow linewidth" and "second-order coherence" linked to "single-photon statistics". First I am not convinced that the linewidths are narrow here (especially compared to the lifetime-limited linewidth). Secondly this first-order coherence is related to single-photon interference, which is to be avoided when implementing HOM (as in the current manuscript, the interferometer optical path difference being much longer than the single photon coherence length). At last, the $g^{(2)}$ function is the second-order auto-correlation function. Here it seems inappropriate to refer to it as a probe for second-order coherence as neither coherence time nor frequency linewidth are extracted. In my opinion, the term "coherent single-photon source" brings confusion here, which tends to distract the readers from the main result (first HOM on a single defect in silicon). I would suggest that the authors clarify or even change this term.

Reply (2.7) We thank the referee for drawing attention to better definitions of coherence. In our experiments, the inhomogeneous broadening is indeed large (2.8 GHz). However, the HOM experiments demonstrate narrower linewidths over short timescales (discussion below).

We agree that one has to be careful while referring to distinctions between second- and first order coherence. Standard antibunching ($g^{(2)}(\tau)$) measurement of an emitter (e.g. Fig. 3(c)) measures the intensity correlations and does not contain information about the first-order coherence. However, we note that the measured $g^{(2)}$ function for the HOM interference does contain information about the first-order coherence as the interference time window is determined by the first-order coherence time. This can be seen in Eqn. (2) in Lettow

$$g_{34}^{(2)}(\tau) = c_1^2 g_{11}^{(2)}(\tau) + c_2^2 g_{22}^{(2)}(\tau) + 2c_1 c_2 \left\{ 1 - \eta \frac{\langle S_1 \rangle \langle S_2 \rangle}{\langle I_1 \rangle \langle I_2 \rangle} g_{11}^{(1)}(\tau) g_{22}^{(1)}(\tau) \cos(\Delta\omega\tau) \right\}$$

where the intensity cross correlation function $g_{34}^{(2)}(\tau)$ depends on the first-order coherences of the photons in

the interfering wavepackets, $g^{(1)}$

$g_{11}^{(1)}(\tau)$ and $g_{22}^{(1)}(\tau)$. Intuitively, the first order coherence of the fields is related to the interference window in an HOM measurement.

However, we agree with the reviewer that "highly coherent" is a subjective statement, and may be confusing or misleading. So, we have removed those words from the abstract. This is the only place in the manuscript where we refer to the G center as 'highly coherent'. We believe that the other uses of coherence in the text are accurate. *Line-indexed revisions:* 11

Referee #3

The authors have not made significant changes to the manuscript in their revision. In particular, a more detailed discussion of the observed indistinguishability is still lacking in the main text. All referees had critical questions on the analysis; while responses are given, only slight changes have been made to the manuscript. Assessing the indistinguishability based solely on $g^{(2)}(0)$ remains incomplete, and the brief treatment of the observed time constant of the quantum interference dip in the main text leaves questions open. One must study the supplemental material to obtain a more detailed analysis and a discussion of this result. It would be helpful to have more of this in the main text. I would have expected a mentioning of the indistinguishability of the entire photon wave packet for comparison, as well as a discussion of how the results compare to the measured spectrum.

Reply (3.1) We agree with the referee that additional detail in the main text would be helpful to readers in interpreting the results. To address this concern:

Revision (3.1) We added the following to the main text: (i) discussion of indistinguishability over the entire wavepacket, (ii) the discussion on the model and quantum interference time constant, and (iii) comparison of this constant with the measured spectrum.

Line-indexed revisions: 102-118

I now also wonder what the effect of the filter FP cavity is, given that it is more narrow than the emitter linewidth – I missed this aspect when reviewing the first version of the manuscript. What indistinguishability would be observable without the filter cavity? A narrow enough spectral filter can produce indistinguishable photons from any single photon emitter, at the cost of low efficiency. How is this trade-off here? While the effective linewidth for quantum interference derived in the supplement suggests that the filter effect of the cavity may be small, an estimate should be given. Overall, I think that these aspects should be addressed before publication.

Reply (3.2) The linewidth of our G center (2.8 GHz) is narrower than the linewidth of our Fabry-Perot cavity (3.4 GHz). The linewidth of the G center appears broader than the linewidth of the filter in Figure 3 because the signal from the G center is convolved with the linewidth of the filter, as clarified in Figure 3's caption and the main text.

Nonetheless, the referee raises an important point, as even though the filter is wider, it has a Lorentzian profile and will alter the effective linewidth of the transmitted photons. A 2.8 GHz source transmitted through a 3.4 GHz filter will have a linewidth of ~ 2 GHz. However, this linewidth is the long-term linewidth taken over several minutes, while we measure a much narrower effective linewidth $\Gamma^{\text{HOM}}/2\pi = 0.4 \pm 0.1$ GHz at the 25 ns timescale, as the referee has pointed out. We estimate a deviation of less than 2% for the measured effective linewidth after transmission through the filter. This deviation is significantly below the error of our reported value.

Revision (3.2) We have clarified that the effect of the FP filter on Γ^{HOM} is negligible.

Line-indexed revisions: 111-112

We again thank all referees for their valuable feedback. Their questions, criticisms, and comments allowed us to strengthen our manuscript, and improve its clarity. We believe we have addressed all of their concerns, and that our revised manuscript meets the high standards for publication in *Nature Communications*.

REVIEWERS' COMMENTS

Reviewer #2 (Remarks to the Author):

I am satisfied with the authors' responses, as well as with the revisions they have made to the manuscript.

I recommend its publication in Nature Communications journal.

I would like to point that these results are the first report of HOM interference from a single defect in silicon, furthermore at telecom wavelengths. Although the use of temporal filtering is not recommended in the long term due to its negative impact on the success rate, it has recently been used to demonstrate 2-photon quantum interference between 2 remote T centers in silicon (Afzal et al., arXiv:2406.01704 (2024)), with significant inhomogeneous broadening as here.

Reviewer #3 (Remarks to the Author):

The authors have addressed all points raised and made according changes to the manuscript. With this I believe that the work is suitable for publication in Nature Communications.